# Quantum-assisted distortion-free audio signal sensing

Chen Zhang[1] ✉, Durga Dasari[1] ✉, Matthias Widmann[1], Jonas Meinel [1], Vadim Vorobyov[1], Polina Kapitanova [2], Elizaveta Nenasheva[3], Kazuo Nakamura [4], Hitoshi Sumiya[5], Shinobu Onoda [6], Junichi Isoya [7] & Jörg Wrachtrup[1]

Quantum sensors are known for their high sensitivity in sensing applications. However, this sensitivity often comes with severe restrictions on other parameters which are also important. Examples are that in measurements of arbitrary signals, limitation in linear dynamic range could introduce distortions in magnitude and phase of the signal. High frequency resolution is another important feature for reconstructing unknown signals. Here, we demonstrate a distortion-free quantum sensing protocol that combines a quantum phase-sensitive detection with heterodyne readout. We present theoretical and experimental investigations using nitrogen-vacancy centers in diamond, showing the capability of reconstructing audio frequency signals with an extended linear dynamic range and high frequency resolution. Melody and speech based signals are used for demonstrating the features. The methods could broaden the horizon for quantum sensors towards applications, e.g. telecommunication in challenging environment, where low-distortion measurements are required at multiple frequency bands within a limited volume.

Quantum sensors are setting new frontiers of sensing techniques with their extraordinary performances in sensitivity and stability[1–5]. These techniques rely on either measuring the line-shift of spin or atomic transition frequencies or reading out the relative populations of the occupied energy levels using interferometric methods[6,7]. In most cases, there are trade-off relations between the sensitivity and other relevant features in metrology[8]. For example, a high-sensitive measurement acquired by detecting the transition line shift requires a narrow linewidth, which, on the other hand, will limit the dynamic range if no feedback is applied. Interferometric measurements detect a sinusoidal response, and linearity is only achieved when the phase signal is in a small dynamic range, which hampers sensing unknown

signals in practice. In order to reconstruct arbitrary signals, extending the linear dynamic range (LDR), resolving signal frequency in high resolution, and maintaining a high sensitivity shall all be simultaneously addressed.

The nitrogen-vacancy (NV) centers have been at the forefront in performing high-sensitive measurements of various physical quantities, viz., magnetic and electric field, temperature, and strain distributions internal and external to diamond[9–14]. The NV magnetometry has been performed under bias fields ranging from zero-field to a few Tesla[15–18]. Compared to optically pumped magnetometer (OPM) which has a high sensitivity at dc but degrades at frequencies larger than a few hundreds Hz[19–21], NV magnetometry is

[1]3rd Institute of Physics, University of Stuttgart, Allmandring 13, 70569 Stuttgart, Germany. [2]Department of Physics and Engineering, ITMO University, Saint Petersburg 197101, Russia. [3]Giricond Research Institute, Ceramics Co. Ltd., Saint Petersburg 194223, Russia. [4]Hydrogen and Carbon Management Technology Section, Hydrogen and Carbon Management Technology Strategy Department, Tokyo Gas Co. Ltd., Yokohama 230-0045, Japan. [5]Advanced Materials Laboratory, Sumitomo Electric Industries Ltd., Itami 664-0016, Japan. [6]Takasaki Advanced Radiation Research Institute, National Institutes for Quantum Science and Technology, Takasaki 370-1292, Japan. [7]Faculty of Pure and Applied Sciences, University of Tsukuba, Tsukuba 305-8573, Japan. ✉ e-mail: c.zhang@pi3.uni-stuttgart.de; d.dasari@pi3.uni-stuttgart.de

capable to detect signals from dc to GHz[22,23]. In NV magnetometry, dynamical-decoupling techniques are usually employed to enhance sensitivity[9,24,25]. High frequency resolution can be achieved with the quantum heterodyne (Q-dyne) detection technique[26,27]. The dynamic range can be extended by using phase-estimation algorithms (PEA)[28,29] and feedback schemes[30,31]. However, no scheme so far has been demonstrated that is capable of achieving both features at the same time.

One of the most fundamental features of a distortion-free sensing scheme is to have a linear response to the external field, which we dub here as readout linearity. Frequency-locking feedback schemes that track the resonance frequency shifts of the optically detected magnetic resonance (ODMR) spectrum is applied to NV magnetometry for extending the dynamic range of dc sensing[30]. However, this method can hardly be used in traditional interferometric schemes for measuring oscillating fields, in which spins interact with the fields and accumulate the quantum phase for readout. PEA can effectively improve the dynamic range for interferometric schemes by using different quantum-phase integration time as resources for the algorithm, in either a non-adaptive or adaptive scheme[28,29,31,32]. The precision of PEA depends on how many resources are used, and the PEA readout also has cross talk between the signal frequency offset and the phase[32].

We note that the distortion-free quantum sensing technique could benefit applications, e.g., wireless communication in challenging environment[33,34]. Conventionally, antennas are the most commonly used radio-frequency sensors for sensing either electric fields or magnetic fields, and both types have achieved very high sensitivity[35–37]. Nevertheless, in challenging environment, e.g., underwater and underground, where low-frequency radio signals are preferred, the conventional antennas should be of the size of (sub)meters depending on the wavelength, thereby limiting their applications. With the recent progress in the field of atomic (Rydberg) sensors, broadband local electric field sensing with wide bandwidth up to 20 GHz has been achieved[38,39]. However, an antenna and a preamplifier are still required due to the low local-field sensitivity, making this waveguide-coupled system less compact. In this regard, quantum magnetometers can be attractive for their high sensitivity and compactness[33,40–42]. NV magnetometers, despite their lower sensitivity when compared to OPMs, have shown the capability for reconstructing signals in multiple frequency bands. Combing NV magnetometers with flux concentrators can make them both sensitive and compact due to the high volume normalized sensitivity (more comparison details see in Supplementary Note 2)[43,44].

In this work, we improve on the feasibility of distortion-free magnetometry with quantum sensors by using NV center ensembles in diamond as the experimental platform at room temperature. Firstly, we introduce the quantum phase-sensitive detection (QPSD) technique, which builds upon the idea of the classical lock-in technique, and provides an extended LDR for interferometry schemes by using two synchronized driving fields with a frequency offset. Then, we present the heterodyne readout, which resolves frequency of the ac signals to be detected. In the scheme, we use a single type of sequence that combines the QPSD technique and the heterodyne readout so that distortion-free sensing can be performed. It is also noticed that the frequency comb induced by continuous sampling can be applied to Hahn-echo sequence for measuring signals beyond the coherence limit without losing sensitivity. We present arbitrary signal measurements at audio bands from 10 to 15 kHz in both time domain and frequency domain. Moreover, a piece of melody and a speech are encoded to a carrier at 10 kHz, and the audios are reconstructed from the readout of the NV magnetometer. By using the sensor as a quantum radio, we demonstrate the application potentials for areas such as quantum-assisted telecommunication and unknown signal exploration.

## Results

### Quantum phase-sensitive detection

Based on the idea of lock-in detection, the QPSD scheme requires an modulation of the quantum phase along with the interferometry of the sensor qubits, so that the phase factor accumulated by the spin-field interaction can be extracted by demodulating the output of the sensor. The recently proposed Q-dyne methods acquire such a modulation by utilizing the phase difference between the qubit and an external oscillator, as shown in Fig. 1a[26,27]. However, the modulation frequency has a dependency to the signals. In order to modulate the quantum phase independently, it is proposed to use a frequency-offset scheme[45,46]. In the scheme, the spin is manipulated under one rotating frame while being observed by another one. The two rotating frames are defined by two driving fields with a frequency offset, which leads to a relative rotation between them, i.e., rotating frame modulation. In this way, the quantum phase modulation only depends on the frequency difference of the two driving fields, as shown in Fig. 1b, c. By performing multiple measurements and using lock-in detection, we extract the quantum phase with linearity over a dynamic range of $[-\pi, \pi]$, which is naturally limited by the $2\pi$ wrapping. Further extending of the magnetic field dynamic range can be achieved by adaptively setting the phase reference in the demodulation or address the wrapping in algorithm. Below we describe the rotating frame modulation in theory.

Aligning an external field $B_0$ along the NV axis, we use the two-level subspace of the NV ground triplet. The Hamiltonian of the system is:

$$\mathcal{H} = \omega_0 S_z + \gamma_e B_1 \cos(2\pi f_1 t + \alpha) S_x, \tag{1}$$

where $\omega_0$ is the transition frequency of the two-level subspace, $B_1 \cos(2\pi f_1 t + \alpha)$ is the oscillating driving field with frequency $f_1$ and phase $\alpha$, and $\gamma_e$ is the electron gyromagnetic ratio. In the rotating frame defined by $\omega_0$, the Hamiltonian is:

$$\mathcal{H}'_1 = \Omega_1 \cos(\delta\omega_1 + \alpha) S_x + \Omega_1 \sin(\delta\omega_1 t + \alpha) S_y, \tag{2}$$

where $\delta\omega_1 = 2\pi(f_0 - f_1), f_0 = \omega_0/2\pi$, and $\Omega_1 = \gamma_e B_1/2$ is the Rabi frequency introduced by MW1. In interferometry measurements, a $\pi/2$ pulse prepares the spin state from the polarized state to an equalized population, and another $\pi/2$ pulse projects the quantum phase as a population difference after the sensing procedure. We use the second driving field, MW2, to offset the frequency of the second $\pi/2$ pulse. $\delta\omega_2, \Omega_2$ and $\beta$ are used to denote the frequency offset, Rabi frequency, and phase of MW2. After this, the measured spin-expectation value is:

$$\langle S_z \rangle = \sin\left[\phi + \frac{\pi}{2}\left(\frac{\delta\omega_1}{\Omega_1} - \frac{\delta\omega_2}{\Omega_2}\right) + \alpha - \beta\right], \tag{3}$$

where $\phi$ is the acquired quantum phase which contains the information we want to measure, both of the MWs are near-resonant with $\delta\omega_1 \ll \Omega_1, \delta\omega_2 \ll \Omega_2$ (derivation see Supplementary Note 3). Therefore, the off-resonant term can be neglected, and the phase difference term $\alpha - \beta$ will evolve with time so that there is:

$$\langle S_z \rangle \approx \sin(\phi + 2\pi\delta f \cdot t), \tag{4}$$

where $\delta f$ is the frequency difference of the two MWs.

The above result can be seen as a modulation of the rotating frame itself. As schematically shown in Fig. 1b (left Bloch sphere), we assume that the two driving fields have the same phase at sample 1, and this defines an instantaneous rotating frame with coordinates $x_1y_1z$. Thus, the readout is similar to that of the regular Ramsey measurement. After an interval of $\Delta t$, MW2 develops a phase difference of $2\pi\delta f\Delta t$. Since the quantum phase is finally measured by MW2, the

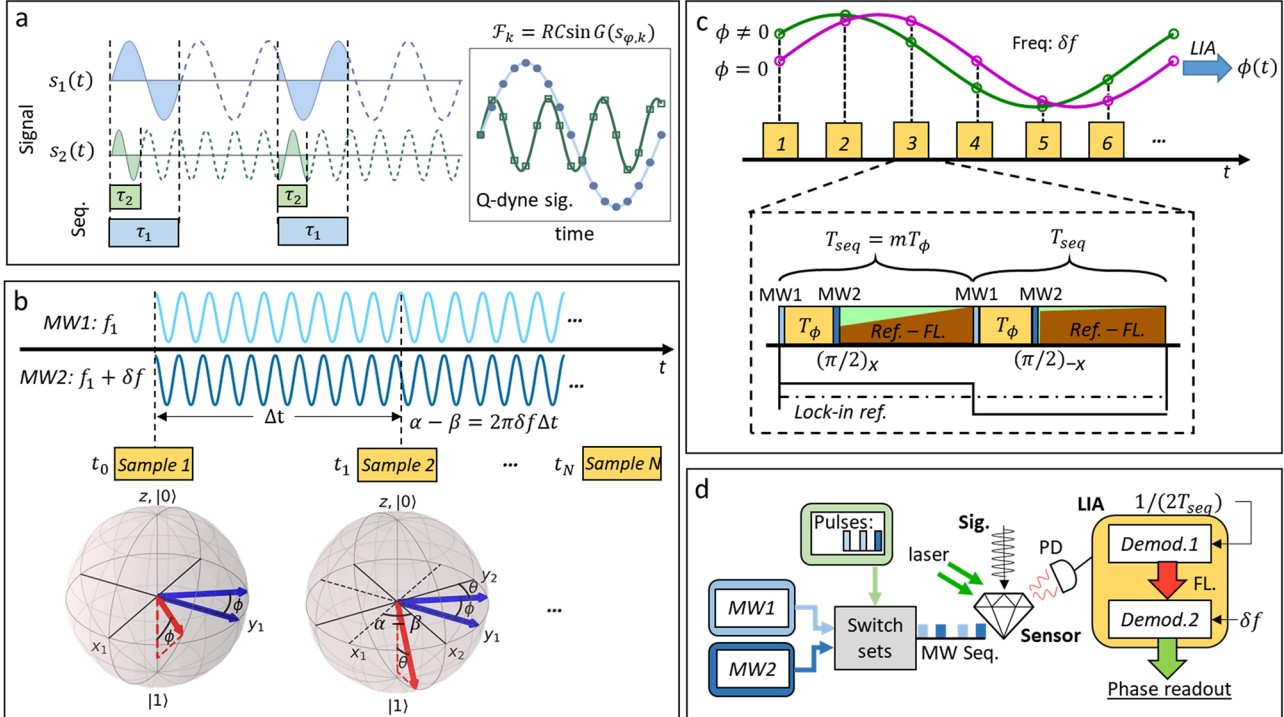

**Fig. 1 | Phase-sensitive NV magnetometry. a** Continuous sampling induced phase reviving signals, known as the quantum heterodyne (Q-dyne) detection. Field interaction time should be set in measurement sequences as $\tau_1$ or $\tau_2$ for different frequencies of the signals $s_1$ or $s_2$. $\mathcal{F}_k$ is the fluorescence signal, where $k$ denotes the number of the output series number, $R$ is the photon rate and $C$ is the contrast. $G(s_\phi, k)$ is the sequence response to the field $s$, which should be small to ensure the linearity. **b** Rotating frame modulation induced by the evolving phase difference of the two driving MW fields. The MWs acquire a phase difference as $\alpha - \beta = 2\pi\delta f\Delta t$ after a sampling time interval $\Delta t$ due to the frequency offset $\delta f$. The picture of Bloch spheres shows the process of rotating frame modulation, in which $x_1 y_1 z$ is the rotating frame defined by MW1, and $x_2 y_2 z$ is the rotating frame defined by MW2. $\phi$ is the accumulated quantum phase during the sensing interval. The acquired quantum phase is $\theta = 2\pi\delta f\Delta t - \phi$ at sample 2. Both the red vectors are the final states of the two samples, and the projections are acquired as the curves in **c**, where we present the measurements when $\phi = 0$ and $\phi \neq 0$. The phase factor $\phi$ can be extracted by a lock-in amplifier. Measurement sequence is shown in the dashed box. $T_\phi$ is the field interaction time, during which pulses are applied using MW1 The light and dark blue blocks are the $\pi/2$ pulses used for the interferometry and the dark blue pulses use MW2. The green and red represent the acquisition windows, in which the green is the laser reference and the red is the fluoresces. $T_{\text{seq}}$ is the length of each sequence part and $m \geq 2$ is an integer. **d** Schematic of the experiment. NV centers ensemble in diamond is used to perform the QPSD readout.

Bloch vector rotates in the new instantaneous rotating frame with coordinates $x_2 y_2 z$, as shown in Fig. 1b (the right Bloch sphere). The rotating frame defined by MW2 rotates continuously around the $z$-axis with the frequency of $\delta f$. Due to this, the fluorescence readout is also modulated in a sinusoidal fashion, as shown in Fig. 1c. By fitting or demodulating the fluorescence signal, we can resolve the changing of the phase factor $\phi$ between each modulation cycle and find linearity to the external field. The measurement sequence we applied in the experiment is depicted in the dash box of Fig. 1c, in which $T_\phi$ is the field sensing time, $T_{\text{seq}} = mT_\phi$ is the sequence length of one measurement, and we use a second measurement with the final pulse changed to $(\pi/2)_{-x}$. The experimental schematic is shown in Fig. 1d (details see in Methods "Experimental setup").

The sensitivity limit is derived based on the fitting of the $N$ samples in the measurement time of $N \cdot 2T_{\text{seq}}$[47]. The minimum detectable phase is:

$$\delta\phi = \frac{2\sqrt{2}}{\sqrt{N}} \frac{1}{Ce^{-(T_\phi/T_2)^p}\sqrt{\mathcal{N}}}, \quad (5)$$

where $C$ is the fluorescence signal contrast, $\mathcal{N}$ is the detected photon counts in each fluorescence readout window, $e^{-(T_\phi/T_2)^p}$ indicates the contrast reduction due to the decoherence time $T_2$ and $p$ is the stretched exponential parameter. In the derivation, there is a factor of 2 due to the two steps measurement and the laser reference. The sensitivity to external magnetic field is also subject to the MW sequence, and can

be derived as:

$$\eta = \frac{4}{\gamma_e|G(\omega)|Ce^{-(T_\phi/T_2)^p}}\sqrt{\frac{T_{\text{seq}}}{\mathcal{N}}}, \quad (6)$$

where $|G(\omega)|$ is the MW filter function which is used to describe the transfer function from magnetic field to quantum phase. In comparison to the fluorescence readout, the sensitivity of QPSD readout deteriorates by a factor of $\sqrt{2}$ (derivation see Supplementary Note 5).

In Fig. 2a, b, we compare the regular interferometry (single driving field) and with the measurements obtained from the QPSD readout described above. Both the readout spectra and the responses to the test fields are plotted in the figures. The strength of the applied external ac fields ranges from 0 to 3 μT. For Ramsey measurements, the applied fields are at a frequency of 46 Hz, and we use a field sensing time $T_{\phi,\text{Ramsey}} = 6.25$ μs. For Hahn-echo measurements, we use external fields at 80.046 kHz and the field sensing time $T_{\phi,\text{Hahn}} = 2T_{\phi,\text{Ramsey}}$. The signal readout of the regular interferometry measurements is proportional to $\sin\phi$, where $\phi \propto \gamma_e B$ is the quantum phase. Thus, the regular Ramsey and Hahn-echo readout quickly saturate because the response is linear only when $\phi$ is small. In the spectra, harmonics of the 46 Hz signal rise significantly due to the saturation induced by the limited LDR, compared to the QPSD readout which shows the linearity over the field range. The high-order harmonics of the signal detected by the QPSD readout are small and mainly arise from the function

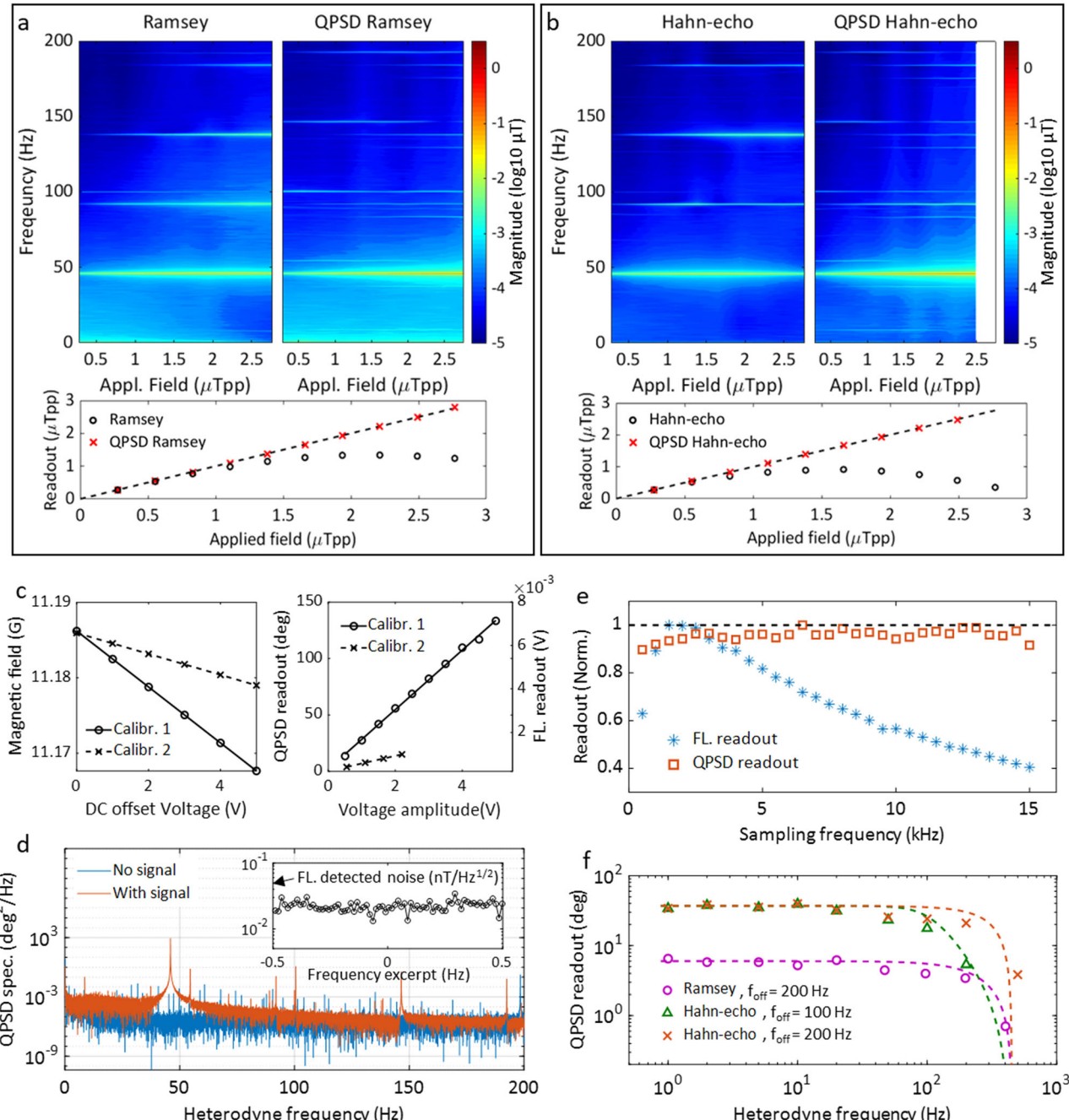

**Fig. 2 | Sensing performance of the QPSD. a** Spectra and linearity comparison of the fluorescence readout and QPSD readout in Ramsey measurements. The measured results of the two readout schemes are both plotted under the spectra, while the dashed line shows the amplitudes of applied fields. **b** Spectra and linearity comparison of the fluorescence readout and QPSD readout in Hahn-echo measurements. **c** Calibration of test coils and the sensor readouts. Calibr. 1 corresponds to the QPSD readout and Calibr. 2 corresponds to the fluorescence readout. The test coil is moved between the two measurements for verification. **d** The noise power spectra of the QPSD readout with/without a calibration signal. The inset is a 1 Hz excerpt of the fluorescence detected magnetic field noise spectrum. **e** Robustness of the QPSD scheme. Normalized signal responses of the two schemes with different sampling frequencies are plotted, and the QPSD readout is expected to be 1 as the dashed line shows. **f** Measurement bandwidth is limited by time constant of the LIA. Ramsey and Hahn-echo sequences are applied to measure test fields at different frequencies with the same magnitude.

generator. In the measurements, one could see the linewidth broadening induced by the increasing signal power. The peak at 100 Hz, which is consistently seen in both the Ramsey and Hahn-echo measurements, comes from the electronics instrumentation. Other side peaks seen near the original signal frequency in the QPSD readout spectra are due to the mixing of the 100 Hz power line harmonics and the 92 Hz signal harmonics in the LIA. It is also noticed that the Hahn-echo measurement meets the phase wrapping when the applied field is larger than 2.5 μT peak-to-peak. We note that additional dynamic range

can be extended by using feedback or algorithms to address the phase wrapping over the cycle of $2\pi$.

The test fields are sent to the diamond by a single loop test coil. The test coil is calibrated by measuring the ODMR spectrum of the sensor, and then, the coefficient is used to calibrate the QPSD readout, as shown in Fig. 2c, d. The acquired fluorescence readout sensitivity is 26 pT/$\sqrt{\text{Hz}}$, while the calibrated QPSD readout sensitivity is 38 pT/$\sqrt{\text{Hz}}$ (details see Methods "Sensitivity estimation"). The technique also demonstrates robustness to changes of $T_{\text{seq}}$. The motivation of

using different $T_{seq}$ is to get different sampling frequencies as well as measurement bandwidths. Signal responses to different sampling frequencies, i.e., $1/2T_{seq}$, are plotted in Fig. 2e. The two sets of values are normalized for comparison because of different units. Ramsey measurements are characterized by a calibration field at 5 Hz. The fluorescence readout shows varying signal responses over the sampling frequency range. The change of the fluorescence contrast is majorly caused by the spin polarization process during the laser readout time[48]. Meanwhile, the QPSD readout is roughly a constant due to the same $T_\phi$. The QPSD readout errors can be attributed to the drifting of the external field over the experimental time. Regardless of sensitivity, the result also indicates that the QPSD readout has no dependency on the fluorescence contrast, which can be affected laser power and the MW power.

The measurement bandwidth of the QPSD readout is shown in Fig. 2f, where the signal responses to different test field frequencies are plotted. The plotted values are the magnitudes at the corresponding frequencies in the Fourier transform of the QPSD readout. For the Hahn-echo measurements, we detected the heterodyne signal for the ac fields. The applied sequence length, $T_{seq} = 100$ μs, gives the referencing frequency $f_s = 5$ kHz for "Demod. 1" (defined in Fig. 1d). The intrinsic fluorescence readout bandwidth is $f_s/2$ according to the Shannon sampling theorem. We apply the second driving field with $\delta f = 500$ Hz to have $N = 10$ samples in a modulation cycle. Due to this, the QPSD bandwidth is narrowed down to $f_s/(2N)$. In addition, the LIA time constant of "Demod. 2" (see Fig. 1d) determines the final bandwidth of the setup with a flexibility $< f_s/(2N)$ (250 Hz in this case). Finally, one can conclude that the rotating frame modulation provides QPSD readout magnetometry that has enhanced LDR and robustness in a flexible bandwidth. As we show below, this can be used for distortion-free arbitrary fields sensing.

## Frequency offset heterodyne readout

Heterodyne readout has been used to improve the frequency resolution in nuclear magnetic resonance spectroscopy. It is also a way to achieve high precision microwave sensing[49–51]. High-order dynamical decoupling sequences are used to narrow the spectral linewidth by decoupling the sensor response from unwanted signal frequencies[26,27]. Here arises a trade-off between the measurable signal bandwidth and fidelity. High-order dynamical decoupling can ensure a high sensitivity but the detectable signal bandwidth is then limited by the filter function defined by the sequence. On the other hand, the lower limit on the detectable signal frequency is set by the decoherence time of the sensor. Here, we will use the Hahn-echo sequence in combination with the QPSD readout to detect signals at frequencies that go beyond the coherence time of the sensor.

In Qdyne, the sampling time usually satisfies $T_{seq} \neq mT_\phi$ so as to get the heterodyne signal[27]. The frequency of this heterodyne signal depends on the timing offset. Here, we choose the measurement sampling time $T_{seq} = mT_\phi$ to obtain the heterodyne readout depending on the signal frequency offset from $1/T_\phi$. As a result, the detected phase of signals at frequencies of $n/T_\phi$ is locked by the sequence, where $n$ can be a random integer. On the other hand, the frequency offset of signals can also introduce phase revivals, i.e., frequency offset heterodyne signal, as shown in Fig. 3a. The detected heterodyne frequency would be the exact offset of the signal frequency to $1/T_\phi$.

The frequency offset heterodyne readout is modeled based on the MW sequence filter[52,53]. Sampling happens in each time interval of

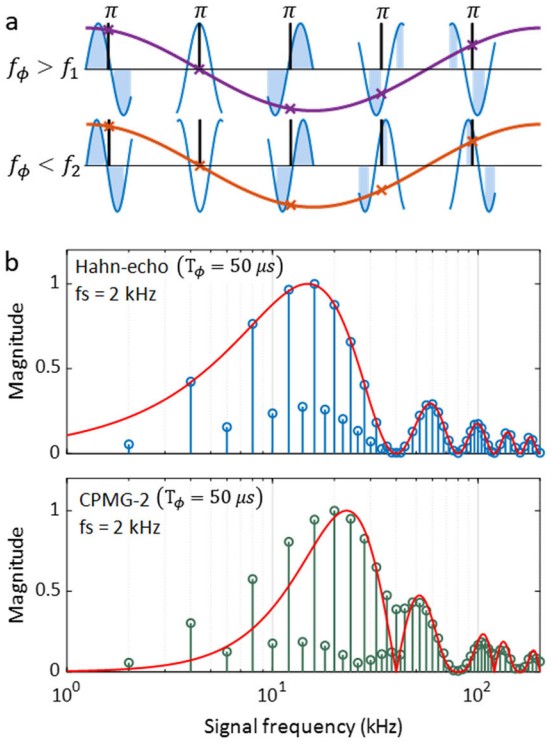

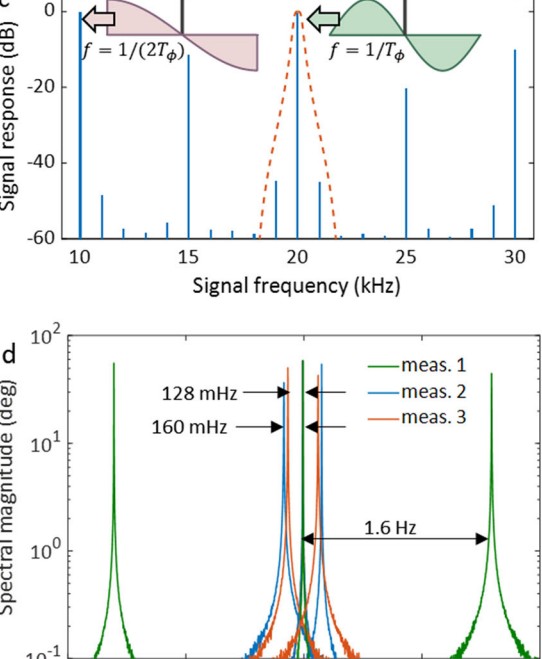

**Fig. 3 | Frequency offset heterodyne readout. a** Principle of the readout scheme. $f_\phi$ is the reference frequency defined by the sequence, and $f_1$, $f_2$ are the frequencies of the signals. The colored regions mark where the quantum phase is accumulated in the Hahn-echo sequence, while phase accumulations at the other areas are canceled in the spin evolution. The figure shows identical heterodyne signals due to $f_\phi - f_1 = f_2 - f_\phi$. **b** Frequency responses of the Hahn-echo sequence and CPMG-2 sequence. The applied ac fields have an 5 Hz offset to the denoted signal frequencies so that they are detected as a 5 Hz heterodyne readout. Both theoretical and experimental results are plotted after normalization. **c** Signal frequency response of Hahn-echo measurements with $1/T_{seq} = 10$ kHz. The dash line indicates the filter introduced by the lock-in amplifier. **d** Heterodyne frequency dependency to sequences and signal. The reference signal is the result of detecting 20.005 kHz signal with sequence that uses $T_\phi = 50$ μs and $T_{seq} = 20T_\phi$. In meas. 1, $T_\phi$ is changed by ±4 ns. In meas. 2, we keep $T_\phi$ unchanged, and offset $T_{seq}$ with ±4 ns. In meas. 3, the applied field is changed to 16.005 kHz while the other parameters are the same as meas. 2.

$[NmT_\phi, (Nm+1)T_\phi]$, where $N \in \mathbb{Z}$, $m \geq 2$ is an integer defined as shown in Fig. 1c. For a random oscillating signal at frequency $\omega$, $B_{ac}(t) = B(\omega)e^{-i[\omega t + \varphi(\omega)]}$, and a measurement with the MW $\pi$-pulse number of $n$, we can get the accumulated quantum phase as (see Supplementary Note 4):

$$\phi_r(N) = |G_n(\omega)|e^{i\left(-\frac{\omega T_\phi}{2} - \frac{P}{2}\pi\right)}\gamma_e B(\omega)e^{-i\varphi(\omega)}e^{-i\omega NmT_\phi}, \qquad (7)$$

where $N$ denotes the sampling timestamp, $G_n(\omega) = |G_n(\omega)|e^{i\left(-\frac{\omega T_\phi}{2} - \frac{P}{2}\pi\right)}$ is the MW filter function, $P = 1$ when the $\pi$-pulse number $n$ is odd and $P = 2$ when $n$ is even. Particularly, when $n = 1$ i.e., Hahn-echo sequence is applied, the filter function satisfies $|G_1(2\pi/T_\phi)| = |G_1(\pi/T_\phi)|$. In principle, measurements of signals at a wide frequency range is feasible by choosing the appropriate $T_\phi$ in Hahn-echo measurements. For example, by using $T_\phi < 1\,\mu s$, one can achieve detection of signals at frequencies higher than 1 MHz. It is more challenging to detect a signal at a lower frequency, such as a signal at 10 kHz, because a longer $T_2$ is required. With the property described above, it is feasible to use $T_\phi = 50\,\mu s$ rather than $T_\phi = 100\,\mu s$ to achieve the measurement with a better sensitivity due to the higher signal contrast when $T_\phi$ is smaller. For diamonds which have NV center ensembles with $T_2 < 100\,\mu s$, the property makes it feasible to detect signals at the frequencies lower than $1/T_2$, i.e., beyond the coherence limit.

Given a reference frequency $\omega_{ref} = k\omega_s, k \in \mathbb{N}$, where $\omega_s = 2\pi/(mT_\phi)$ and $\omega \in \left(\omega_{ref} - \omega_s/2, \omega_{ref} + \omega_s/2\right)$, the evolving phase factor can be rewritten as $e^{-i\omega NmT_\phi} = e^{-i\omega_H t}\delta(t - NT_s)$, where $\omega_H = \omega - \omega_{ref}$ is the heterodyne frequency, $\delta(t)$ is the Dirac function, and $T_s = mT_\phi$ is the sampling period. Thus, the readout signal turns out to be:

$$\phi_r(t) = G(\omega)\sum_{N=-\infty}^{\infty}\gamma_e B_H(t)\delta(t - NT_s), \qquad (8)$$

where $B_H(t) = B(\omega)e^{-i(\omega_H t + \varphi)}$ is the heterodyne readout that contains all the information from the origin signal to be detected. As discussed in previous section, the quantum phase readout bandwidth is limited by the cut-off frequency $f_c$ of the filter of LIA. For any signal with a frequency range of $[(k-1)f_s + f_c, (k+1)f_s - f_c]$, aliasing can be filtered. Although a smaller $f_c$ makes the measurement bandwidth narrower, it ensures signals that in a larger frequency range can be detected without aliasing. By changing $T_\phi$ together with $T_{seq}$, we can resolve a spectrum in multiple frequency bands with a series of sequences.

We present two specific examples of the measured frequency responses by using the Hahn-echo and CPMG-2 sequence. We plot both the theoretical MW filter function and the experimentally detected signal responses together in Fig. 3b. The field sensing time for both experiment and theory calculations is set to be $T_\phi = 50\,\mu s$. The discrepancy at low frequency of the CPMG-2 measurement could be attributed to pulse errors in the experiment. We measured the amplitudes of the frequency offset heterodyne signals with $T_{seq} = 250\,\mu s$, i.e., the magnetic field sampling rate is 4 kHz. Due to this reason, the measured MW filters are combed with a frequency distance of 4 kHz. Aliasing signals exist between the main lobes at a distance of 2 kHz, because the readout sampling frequency is $f_s = 2$ kHz.

In order to detect signals that are distributed over a larger bandwidth, we can increase the sampling frequency, for example, to $f_s = 5$ kHz. The spectrum is plotted in Fig. 3c in decibel, from which one can see that magnitudes are the same at 10 and 20 kHz, i.e., $1/(2T_\phi)$ and $1/T_\phi$ as discussed in the derivation. The insets of Fig. 3c depict the signals that the quantum sensor detects during $T_\phi$ at the two frequencies. In this measurement, the bandwidth limited by the filter of the LIA is at 200 Hz, i.e., the single measurement bandwidth is 400 Hz, and the detectable signal frequency range is 9600 Hz.

We notice that a single measurement cannot tell if the ac field frequency offset is positive or negative from the heterodyne readout. Additional measurements are needed to distinguish the direction of the frequency offset. By adding a difference to the phase accumulation time $T_\phi$ as well as the sequence time, we can change the reference frequency $\omega_{ref}$ to get a different heterodyne frequency. By seeing if the heterodyne frequency increases or decreases, we can determine if the signal frequency is larger or smaller than the reference frequency. As the measurements presented in Fig. 3d that $T_\phi = 50\,\mu s$ is offset by a difference of 4 ns and $T_{seq} = 10T_\phi$ changes accordingly, the detected heterodyne frequency of the signal shift in two different directions. We further investigated the dependency of the heterodyne frequency on the parameters by performing measurements that vary (1) $T_{seq}$, (2) $T_{seq}$ and $\omega_{ref}$. When $T_\phi$ keeps unchanged, the heterodyne frequency shifts by:

$$\Delta\omega_H = \omega_{ref}\Delta T_{seq}/T_{seq}. \qquad (9)$$

Using the equation, we can estimate the frequency fidelity of the given sequence. For example, with a timing error <3 ps, the frequency error of a signal around 10 kHz reduces to 0.06 mHz. The estimation of frequency in spectroscopy is very sensitive to the measurement time $t$. The Cramer-Rao lower bound for frequency measurements decreases as $t^{-3/2}$. A higher frequency resolution can be obtained by increasing the measurement time[26,27,54].

## Sensing of arbitrary audio signals

Taking advantages from the high dynamic range with the QPSD scheme and the frequency resolving from the heterodyne detection, we demonstrate below distortion-free measurements of arbitrary magnetic fields at audio frequencies. The audio frequency band is hard to be covered by the vapor cell based quantum magnetometers[19]. We first generate a signal at 20.08 kHz with its phase varying with time (see Fig. 4a). The MW filter is set by the Hahn-echo sequence with $T_\phi = 50\,\mu s$. With the reference frequency at 20 kHz, the heterodyne readout is at 80 Hz, as seen from the simulated curve. The phase of the external field is switched with a cycle of 80 and 40 Hz so that the experimental readout displays the phase change, as shown in Fig. 4a.

Next, we apply a field with the frequency, amplitude and phase all arbitrarily changing. The signal frequency is around 10 kHz and the signal bandwidth is within 400 Hz. Using $1/T_{seq} = 10$ kHz, we can detect signals close to 10 kHz with the same sensitivity as the 20 kHz signal. The signal length is 1 s and consists of ten 100 ms parts. In Fig. 4b, both the applied field waveform and the QPSD readout are plotted. The heterodyne frequencies can resolve the frequency differences in the original waveform. The amplitudes of the readout also correspond to dynamics of the applied fields.

As discussed previously, the measurement bandwidth used in the experiment is 400 Hz. For this, we perform a spectrum analysis as shown in Fig. 4c. The signal to be detected is a sum of 20 tones with random frequencies, amplitudes and phases. In order to distinguish the sign of frequency offsets for each component, we analyze the signal using an alternative sequence with $T_\phi = 50\,\mu s + 2$ ns. The sharp peaks observed in the spectrum should shift according to the changes of the measurement sequence, else we exclude them as noise signals generated from our electronics. The spectrum analysis algorithm is described with more details in the Supplementary (see Supplementary Note 8). As shown in Fig. 4c, the applied frequencies are properly resolved. Additionally, we find a 9.93 kHz noise spike from the environment. The errors in magnitude of the detected spectrum could be induced by the LIA filter, as shown earlier in Fig. 2f. The errors could also be caused by an insufficient sampling number for demodulating the rotating frame modulation. In the measurements, we apply sequences with their lengths corresponding to a sampling frequency $f_s = 5$ kHz. The frequency difference of the two MWs is $\delta f = 500$ Hz and $N = 10$ for reading out a phase sample. The precision can be increased by using a smaller $\delta f$, but it will decrease the bandwidth.

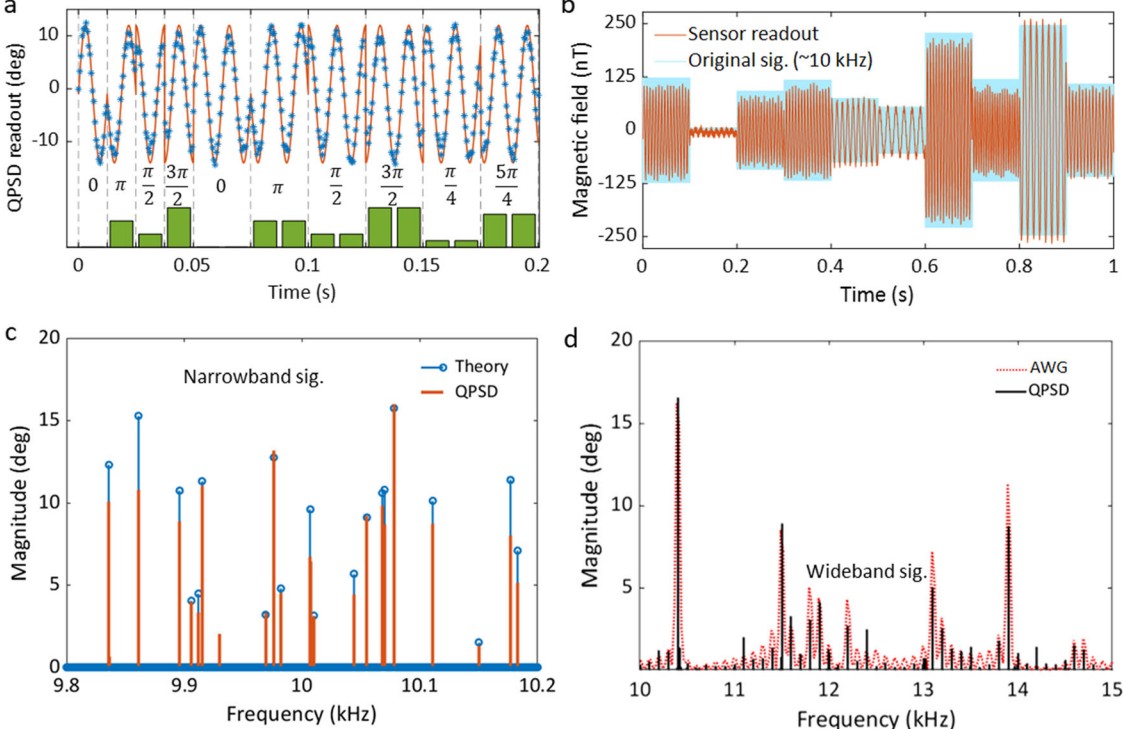

**Fig. 4 | Detection of arbitrary audio signals. a** Phase response of the QPSD measurement. The bars show the phases dynamics of the applied magnetic field. Stars mark the QPSD readout of the sensor, and the curve is the simulated readout. **b** QPSD readout of an ac field around 10 kHz with the frequency, amplitude and phase switched every 100 ms. The waveform of the applied field is plotted in light blue, and the red curve is the QPSD heterodyne readout. **c** Spectral comparison of the applied signal and the detected magnetic field in a 400 Hz bandwidth. The blue remarks denote the applied signals mathematically. **d** A signal with wide bandwidth between 10 to 15 kHz is applied and detected by varying the sequence. The red dash line shows the spectrum of the output of the AWG. The solid black line is the spectrum of the QPSD readout. The magnitude of the AWG spectrum is scaled to the same level of QPSD readout for eye guidance.

Though the bandwidth of each measurement sequence is limited, we can still analyze the signal spectrum within a wider bandwidth by merging several measurements. The condition is that the signal bandwidth should not be larger than the sampling frequency to avoid frequency aliasing. In Fig. 4d, we perform a spectrum analysis for a signal within a bandwidth from 10 to 15 kHz. We set an 800 Hz bandwidth for the measurement of each sequence and use six measurements to cover the entire bandwidth. The signal to be detected is a sum of ten components with their frequencies randomly distributed in the bandwidth. The signal is generated by an arbitrary signal generator (AWG) and sent to the test field coil. The dotted curve in the figure displays the spectrum of the electrical signal from the AWG. There are some harmonics near each main component due to the limited AWG internal clock and signal length. The components at different frequencies are analyzed by varying $T_\phi$ to get different referencing frequencies for heterodyne detection.

Finally, we demonstrate two examples of the distortion-free measurements of modulated analog audio signals. The two reconstructed audios are (i) a melody piece composed of three tones and (ii) the famous "I have a dream" from Dr. Martin Luther King Jr's speech. The reconstructed audios can be heard and compared to the original sounds (see Supplementary Audio 1–4). Although the NV sensor presented in this work has a wide detectable frequency range, the bandwidth determined by the LIA leads to difficulty in detecting real-time signals. The speech signal in case (ii) covers a frequency range from near dc to a few kHz. Therefore, the audio needs to be pre-processed for detection with the sensor, and the readout is post-processed for waveform reconstruction (see Methods "Signal processing for audio sensing"). Nevertheless, the technique is possible to be applied in low-frequency telecommunication for challenging environment, where time lag is not very critical. It is also common that nowadays

information is digitized for broadcasting and receiving, and the technique ensures feasibility of modulating such signals in amplitude, frequency and phase, for the telecommunication applications.

## Discussions

In this work, we overcome the LDR limitation of the conventional interferometric readout through a technique that includes a QPSD scheme and the frequency offset heterodyne readout. The technique allows one to detect unknown signals with maximal sensitivity independent of their dynamic range. It improves the feasibility for not only NV magnetometer, but also other qubit sensors that use quantum coherence, to perform measurements of different physical quantities with high dynamic range and high sensitivity.

Theoretically, the extended LDR comes from the multiple measurements that have the quantum phase evolving through the entire phase range $[-\pi, \pi]$ so that the initial phase factor that contains the external field information can be resolved. Such an extended phase range affects the measurement bandwidth as well as the sensitivity. In theory, the sensitivity does not deteriorate a lot from the conventional fluorescence readout except for a factor of $\sqrt{2}$. We suffer from a low contrast $C < 0.2\%$ due to the low excitation laser power (80 mW) and acquisition with lock-in amplifier (LIA) (see supplements of ref. 48). The contrast and the fluorescence photon count can significantly increase when the laser reaches saturation power. Different dynamical decoupling sequences can also improve the magnetic field sensitivity through the filter function $G(\omega)$. In this work, we measure signals from 10 to 20 kHz and modulate audio signals with such a carrier for the demonstration. However, with the long coherence time $T_2 = 200 \, \mu s$ of the NV ensemble that we use[12], it is possible that we directly measure audio signals at a few kHz by trading the sensitivity. Flux concentration could further improve the signal-noise ratio at a cost of spatial

resolution and the vector property, similar to antennas in conventional radio receiving systems[43,44]. The flux concentrator can be very small (a few centimeters) for potential applications compared to conventional dipole antennas, because the gain no longer depends on the signal wavelength. With the millimeter size diamond dimension, the flux concentration factor can easily reach a factor of hundreds when using a concentrator in centimeters.

The QPSD readout can also enhance the capability of vector magnetometry. Conventionally, fluorescence emitted from NV centers in multiple orientations is measured sequentially to acquire the vector components. Similar to the methods developed here, one could also modulate the signal on each orientation with different modulation frequencies[55]. Performing measurements on different NV orientations with appropriate synchronization can suppress the phase errors in vector reconstruction.

In conclusion, we demonstrated high-sensitive distortion-free quantum-assisted detection of audio signals, including melody and speech, using the QPSD scheme in combination with the heterodyne readout. One could also generalize the current methods to achieve vector magnetometry with extended LDR. We envisage that the techniques developed here lay a path for different quantum sensing platforms in achieving low-distortion sensing required for many applications in science and technology.

## Methods

### Experimental setup

The diamond used in the experiment is a (111)-oriented $(0.5\,\text{mm})^3$ cube obtained from a single crystal grown by the temperature gradient method at high-pressure high-temperature conditions. The diamond is 99.97%$^{12}$C enriched, and has an initial nitrogen concentration of 1.4 ppm. The final NV concentration is 0.4 ppm after electron irradiation and annealing. Dephasing time of the NV ensemble is obtained as $T_2^* = 8.5\,\mu s$ by Ramsey sequence, and a decoherence time $T_2 = 200\,\mu s$ is measured by Hahn-echo sequence. The diamond is positioned at center of a home-built three dimension coils system, which is used to generate a bias field along the (111) orientation. We use a low noise 532 nm laser (Lighthouse Sprout-G) to illuminate the diamond with a power around 80 mW. Fluorescence is collected by a compound parabolic lens and detected by a large area photodiode (Hamamatsu S3590-09). Microwave signals are generated from two sources (Rohde&Schwarz, SMIQ03B) and are individually controlled by two switches. Measurement sequences are generated by a data timing generator (Tektronix, DTG5274). After the combination and amplification of the MW signals, MW pulses are fed to the diamond through a dielectric resonator antenna[56]. The magnetic component of the MW is coupled vertically to the (111) orientation of the diamond. The detected fluorescence signal is demodulated by a LIA (Zurich Instruments, HF2LI) which has two independent differential input channels and demodulators. As shown in Fig. 1d, the fluorescence signal is firstly demodulated at reference frequency of $f_s = 1/(2T_{seq})$, which is also the sampling frequency of the fluorescence readout. The readout of "Demod. 1" is further demodulated by another demodulator of the LIA at the reference frequency defined by the rotating frame modulation, denoted as "Demod. 2".

To generate arbitrary magnetic fields, we encoded signals via an AWG (Tektronix, AWG520) with $10^5$ samples per second output sampling rate. The test signals are continuously repeated and sent to a test coil near the diamond. The test coil is a single round copper loop which is connected with a 50 Ω resistor for the correct loading to signal generators.

### Sensitivity estimation

Sensitivity of the measurements is calibrated by the mentioned test coil, which is driven by a signal generator. The test coil is firstly calibrated by measuring the ODMR spectrum of the NV magnetometer, of which the lines splitting depends only on external field. After calibration, the test coil is used to generate calibration fields to determine the specifications of the other measurement schemes. We implant calibrations based on Hahn-echo measurements used in Fig. 2b, and results are shown in Fig. 2c. The two calibrations corresponds to QPSD readout and conventional fluorescence readout, respectively. We note that the test coil was moved to generate smaller fields that would not exceed the dynamic range of conventional Hahn-echo measurement. From the calibrations, test coil coefficients are calculated through the fitting as $k_{coil,1} = 371$ nT/V, $k_{coil,2} = 139$ nT/V, and the readout coefficients are $k_{r,phase} = 26.33°$/V, $k_{r,fl} = 3.87 \times 10^{-4}$ V/V. Thus, the scalar factors of the two readout methods can be calculated as $k_{phase} = 0.071°$/nT and $k_{fl} = 2.79 \times 10^{-6}$ V/nT. Then, the noise spectra of the QPSD readout and the fluorescence readout are measured. In Fig. 2d, the phase noise power spectra of QPSD measurements with and without a calibration signal (at 80.046 kHz) are compared. There are also noise spikes that the sensor receives from the environment. Excerpts without noise spikes are chosen to evaluate the noise level, and the phase noise level is obtained as $\eta_{phase} = 0.0027°/\sqrt{\text{Hz}}$. Therefore, the magnetic field sensitivity measured by QPSD readout can be calculated as $\eta = \eta_{phase}/k_{phase} = 38$ pT/$\sqrt{\text{Hz}}$. Similarly, the magnetic field sensitivity of fluorescence readout is evaluated as the inset of Fig. 2d shows. The fluorescence readout sensitivity is $\eta_{fl} = 26$ pT/$\sqrt{\text{Hz}}$. Noticing $\eta \approx \sqrt{2}\eta_{fl}$, we confirm the reduction in sensitivity of $\sqrt{2}$ times in the experiment. Based on the phase noise of the QPSD readout, we further calculate the dynamic range without addressing the phase wrapping. Since the largest amplitude of an ac signal is limited by $\pi$ within the range of $[-\pi, \pi]$, the dynamic range is calculated as $DR = 20\log(\pi/\eta_{phase}) = 96.5$dB. For measurements using a dc sensing scheme, i.e., Ramsey sequence, the largest signal is limited by $2\pi$, and the dynamic range increases by 6 dB.

### Spectrum analysis

The spectrum to be analyzed is divided into several sections with the bandwidth set by the LIA for data acquisition. In each section, the center frequency determines $T_\phi$ of the measurement sequence. Usually, the center frequency satisfies $f_c = 1/T_\phi + \varepsilon/T_{seq}$, where $\varepsilon = 0, \pm 1$. A time trace is recorded after running the sequence, and a spectrum is acquired from the Fourier transform of the time trace. However, the spectrum is a fold of the two sidebands with respect to the center frequency. The sequence with $T' = T_\phi + t_{clk}$ and $T' = mT'$ is applied to get an alternate spectrum with analyzed frequencies shift by $\Delta f = \pm|1/T_\phi - 1/T'|$. The direction of the frequency shift shows which side the signal component belongs to. In the algorithm, we set a threshold to separate signal spikes from noise, and use the known sequences induced spectrum frequency shift to distinguish the signs of the signal offset frequency to the center frequency. The signal spikes that do not shift according $\Delta f$ are recognized as systematic noise spikes. Then, the spectrum of the selected section can be replotted as the example shown in Fig. 4c (more information see Supplementary Note 8). After measuring the spectra of all the sections, we can get the final spectrum by merging them together.

### Signal processing for audio sensing

In the first audio sensing example, the tones of the melody have frequencies distributed between 500 and 700 Hz, of which the frequency range is within the bandwidth of the sensor. In order to modulate the signal to a detectable frequency range of the sensor, the analog audio signal is mixed with a 9.5 kHz reference so that the magnetic signal for detection is around 10 kHz. The heterodyne readout of the NV magnetometer itself plays as a demodulation of the carrier. After the detection, the readout is mixed with the 500 Hz base as the postprocessing for reconstructing tones of the melody. For case (ii), the original audio signal covers a frequency range from near dc to a few kHz, which is larger than the bandwidth of the sensor. Therefore, we compress the signal bandwidth into 200 Hz by interpolation, and then

mix the signal with a 10 kHz reference. Similar to the reconstruction of the melody signal, the speech is demodulated by the heterodyne readout. As the post-processing step, the readout is expanded in frequency domain by averaging the readout in time domain. In Supplementary Note 7, we provide more details about the signals.

## Data availability
All data generated in this study have been deposited in DaRUS under accession with https://doi.org/10.18419/darus-2793.

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

## Acknowledgements

We acknowledge financial support by European Union's Horizon 2020 research and innovation program ASTERIQS under grant No. 820394 (J.W.), European Research Council advanced grant No. 742610, SMel (J.W.), Federal Ministry of Education and Research (BMBF) project MiLi-Quant (J.W.) and Quamapolis (J.W.) and Cluster4Future-QSens (J.W.), and Japan Society for the Promotion of Science (JSPS) KAKENHI No. 17H02751 (J.I.) and No. 20H00340 (J.I.).

## Author contributions

C.Z. wrote the original draft with the assistance of D.D. All authors helped review and edit the article. C.Z. performed the experiments with discussion and assistance from M.W., V.V., J.M. and J.W. The diamond material is supported from K.N., H.S., S.O. and J.I. The microwave dielectric resonator is supported from P.K. and E.N. The work is supervised by J.W.

## Funding

## Competing interests

The authors declare no competing interests.
