## [Peer Review File · Nature Communications]

Quantum-assisted distortion-free audio signal sensingREVIEWER COMMENTS

Reviewer #1 (Remarks to the Author):

In this work, the authors overcome the limitation in linear dynamic range for quantum sensors, in particular for NV-based magnetometers. The proposed scheme is quite novel and interesting, which combines the advantage of various quantum sensing strategies, achieving high measurement sensitivity, large linear dynamical range, and high-frequency resolution (as a remark: I am not sure whether "arbitrary frequency resolution" is very precise because I believe that there shall be some limits anyway. Similarly, I feel a bit uncomfortable about statements like "to measure <arbitrary> signal"). Perhaps, the authors can be more precise in these statements.

The authors point out the advantage of their scheme with electric-field sensors in the small sensing volume, such as Rydberg atom sensors. It would be nice to explicitly compare their sensitivities as one may wonder the sensing volume (the dimension of antennas) may not be very critical when considering applications in telecommunications.

Another concern related to the real application is how the performance of the scheme will be for signals of different frequencies, e.g. in the range of 100MHz - 10 GHz.

Overall, I feel that the work would advance the field significantly and would be happy to recommend for its publication.

Reviewer #2 (Remarks to the Author):

This work uses Nitrogen-Vacancy (NV) ensemble probe to demonstrate low frequency ~ 10 kHz ac magnetometry that can quantitatively sense not only the amplitude but also phase of the signal. This work extends the Qdyne technique presented in Ref[20] by essentially modulating the rotating-frame and thereby obtain the phase information. Demonstration with audio signals is interesting. For the demonstration with the speech, the original audio signal is slowed down by 20 times to facilitate sensing using the NV probe. Regardless, this is an interesting work. I only have minor comments and otherwise the manuscript is well suited to be published in Nature Communications.

Authors make good reference to adaptive/nonadaptive schemes for quantum phase measurement with improved LDR. Note that while Ref[24] is an adaptive scheme, Ref[23] uses a nonadaptive scheme so the sentence in 3-52 should be corrected accordingly.

I would like to bring authors attention to another paper where nonadaptive quantum phase estimation scheme was extended to detect ac magnetic fields:

<https://doi.org/10.1103/PhysRevB.88.220410>

As the above work also demonstrate ac magnetometry with quantum phase sensing with improved LDR, it is indeed relevant to this manuscript so better cite it for completeness.

Fig1(b) caption should be corrected to "..it can be understood.." .

Fig4(b) captions says "..light blue corresponding to the right y-axis.." but there is no right y-axis.

Fig4(d) better have legends to indicate what what red and black curves are.

18-377 should be corrected to ".. pulses are fed .."

One important data processing step for recognizing the systematic noise is explained in the end of Page 18 (18-392 onwards). It would be informative to see, perhaps in the supplementary, how the spectra looks like before and after applying this filtering method.

Reviewer #3 (Remarks to the Author):

The Manuscript, "Quantum-assisted distortion-free audio signal sensing" introduces a general technique to increase the linearity of a sensing pulse sequence while maintaining bandwidth and sensitivity. The quantum phase sensitive detection (QPSD) technique operates by applying two modulating MW fields to the NV ensemble which, after demodulation, allows for the sensor to cover the full linear dynamic range of the phase-sensitive measurement. The manuscript demonstrates a QPSD-modified Ramsey and Hahn-Echo sequence, showing potential to detect a larger range of magnetic field amplitudes than pure implementation of Ramsey and Hahn-Echo. The sensitivity achieved by the sensing protocol is degraded by a factor of $\sqrt{2}$, with each sample taking two steps to acquire instead of one. This technique builds on a combination of existing techniques, and to the reviewer's knowledge, the combination has not been applied to diamond. Substantial revision addressing advantages of this technique with respect to existing methods, substantiation of claims of what has been achieved, clarity in description such that the work could be reproduced, and general precision in wording could warrant publication in Nature Communications.

Audio sensing:

While the increase in dynamic range technique is interesting and useful to the field, the central message of the paper of audio field sensing as presented is misleading. The paper and supplementary files give the impression of actual audio sensing, while audio field signals were never measured in real time. The audio signal was classically preprocessed (compressed from 4 kHz to 200 Hz) and subsequently broadcast at this lower bandwidth in order to make the signal detectable by the NV diamond sensor. The NV-diamond-received-signal needed to be post processed (averaging sequential points in the time domain) in order to reconstruct an audio signal. The comparison of audio measurement to existing techniques is quite minimal, starting on line 58. This description incorrectly portrays quantum-based Rydberg sensors as requiring an antenna. Since they do not require an antenna, a comparison of sensitivity of the two, perhaps for a propagating EM field would be useful. Similarly, this could be extended to antenna. A stronger point, which would need analysis, could be that Rydberg sensors could also benefit from the QPSD higher dynamic range and extended bandwidth.

Dynamic range:

As dynamic range is a significant point of the manuscript, an intuitive description (beyond lines 128-130) of how this technique works would be helpful. In addition, a quantitative method comparison table of the traditional (Ramsey, frequency locking, other techniques) vs. QPSD technique is needed. This table could show trades in sensitivity, dynamic range, frequency range of applicability, for example, putting the work into context. Figure 2 shows an experimental comparison of QPSD to Ramsey sensing sequences. Significantly, what the limit of QPSD, theoretically? The manuscript seems to be motivated by practicality of pulse sequence for increasing dynamic range and bandwidth. Translating $-\pi$ to π to frequency could potentially assist in relaying this description. Is figure 2C referring to Ramsey readout out? These plots should be compared theoretical curves.

Substantiation of claims:

In general, throughout the paper, the claims need to be substantiated. The abstract notes experimental and theoretical investigations reaching 98 dB linear dynamic range, 31 pT/rHz sensitivity, and arbitrary frequency/phase resolution (which would be limited by a master clock). None of these numbers seem to be substantiated in the manuscript. The sensitivity value is stated and the dynamic range is inferred from scale factor. Line 315 notes that a sensitivity is achieved, but there is no data to show how this is obtained. The manuscript notes that the experiment is performed under low contrast (how low?), and under these conditions, it would be difficult/impossible to reach shot noise limit. Details on experimental parameters are needed. Does this "achieved" sensitivity agree with the theoretical prediction of $\sqrt{2}$ less than traditional methods? Give sensitivity for both Ramsey and Spin Echo with and without QPSD? Similarly, Line 316 denotes the dynamic range without corroboration. How does the achieved compare with existing techniques? The dynamic range appears to be projected rather than measured. Line 25 says that a narrow NV linewidth limits the full dynamic range of a sensor. This

is only true for open-loop sensor configurations in which the resonance is not tracked. The authors clarify this point later in the manuscript, it should be noted the first time the technique is mentioned. There is also a note that the frequency tracking method is only valid when the frequency to be sensed is less than sampling frequency (line 50). While it does not affect the proof-of-principle result, Figure 2 data are taken in a regime with 46 Hz signals with kHz sampling rates.

Also, the tie to telecommunications is not clear. In addition, the language in this section is not precise, with exaggerated statements. Line 27 notes, "massive limitation on sensitivity". And lines 31-32 note that operating within linear dynamic range of a sensor can be crucial for reconstructing unknown signals, whereas many antenna today have limited dynamic range and solve this issue through post processing and calibration (although it would be better not to have to do so).

Experimental methods:

The experimental methods lack details throughout on how to reproduce the results. To note a few experimental gaps:

- How scale factor is determined/verified to apparatus description including light collection/readout.
- While laser type is given, Sprout G (nominally 10 W) is limited to 80 mW at the diamond.
- Microwaves are delivered through a dielectric resonator without details of delivery, coupling, NV axis orientation; does the resonator effect the bandwidth of signals?
- If test fields are applied by a loop, how does the polarization of the applied test field interact with the NV class being used?
- How is the florescence signal detected?
- The diamond is surrounded by 3D coils. Are they aligned such that all NV classes are used or one is separated out?
- How are the magnetic fields of the test loop calibrated?

Significant points that do not fit within general themes:

Abstract leads with metrology and sensitivity, when neither is the focus of the paper.

Flux concentrators direct more signal at the diamond. This does not fundamentally change the sensitivity of the device, it directs more signal toward the sensing medium at the expense of long term stability, vector accuracy, etc.

NV capability is used interchangeably with quantum sensors. Note that to date sensors such as SQUIDS and vapor cells have far exceeded the sensitivity of NV diamond. NV sensors are not competitive with these devices in the manner portrayed.

Audio signal could be directly sampled by the NV sensor with no loss in sensitivity with a long-coherence-time diamond.

Language tends to overstate without substantiation ex. "massive", "crucial", "remarkably" (in Results section B).

There are quite a few typos in the manuscript: (e.g. Tekreonix on line 376 or frequency offset, phase of MW2, and Rabi frequency, respectively on line 110 is in wrong order). Many variables, lines in plots, etc are not defined. Figure1c: In the pulse sequence diagram, what do the green and brown colors signify? Much of the information in figure descriptions could be in the text itself. Statements such as in line 41 "The NV magnetometry has been performed under bias fields ranging from zero-field to a few Tesla" need citation.

For equation (3), the authors should point to Supplementary note 1 since that is where the expression is derived.

Also, in S3, the evolution operator acts on the wavefunction, not the other way around. The exponential and ket should be switched. Why does the theoretical filter function for the CPMG-2 sequence not really match the measurement? It does for the Hahn-echo sequence.

Point-by-point response

Reviewer #1:

Thank you very much for the comments and recommendation for publication! We are glad to provide response your concerns point-by-point as detailed below.

#1-1: as a remark: I am not sure whether “arbitrary frequency resolution” is very precise because I believe that there shall be some limits anyway. Similarly, I feel a bit uncomfortable about statements like “to measure signal”). Perhaps, the authors can be more precise in these statements.

R1-1: Thank you for the comment. While we inherit the phrase “arbitrary frequency resolution” in heterodyne detection from Ref [24] and [25], we agree that “high frequency resolution” would suit better to avoid any confusion. It is true that the frequency resolution comes with the cost of an increased measurement time, i.e., the frequency resolution is limited by the finite measurement time and hence we do not have arbitrary resolution as the referee correctly pointed. We have also corrected other unclear statements.

So, we have now corrected the statements in **abstract, line 43-44**, add a brief discussion in **line 277-279** with **Ref. [53]**. We have also revised the sentences with “to measure signal” highlighted in the text.

#1-2: The authors point out the advantage of their scheme with electric-field sensors in the small sensing volume, such as Rydberg atom sensors. It would be nice to explicitly compare their sensitivities as one may wonder the sensing volume (the dimension of antennas) may not be very critical when considering applications in telecommunications.

R1-2: Thank you for the valuable suggestion of giving an explicit comparison between the different sensors. To elaborate the discussion, it is definitely a good idea to make a comparison between the sensors. We detailed this discussion by the added **Supplementary Note 2**. In the note, different techniques are discussed and compared in the text with the **Table S2**.

Regarding the arguments of importance in sensing volume, our motivation is to foresee the potential of quantum sensors being developed for a small size system that can benefit the field of telecommunications in challenging environment. For example, under water and underground wireless communication highly rely on low frequency radio signal which has less attenuation than high frequency signal. Conventional antennas for such low frequency signals are not compact (have a meters or sub-meter dimension), which can be a limitation when used in applications with restricted space, e.g. unmanned underwater vehicles. We believe that having small (centimeter-sized), room temperature, high-sensitive sensors for radio signals could benefit many applications. We have clarified and revised the paragraph from **line 62-77** in the **Introduction**.

#1-3: Another concern related to the real application is how the performance of the scheme will be for signals of different frequencies, e.g. in the range of 100MHz - 10 GHz.

R1-3: Thank you for the comment about concerning of the high frequency performance of the sensor. It is a very interesting research direction that several works were published recently discussing high frequency signal sensing using NV sensors (see Ref. [44], [45]). So far, the Q-dyne technique has been performed for microwave (MW) sensing with single NV sensor, at a sensitivity of $\text{nT}/\text{Hz}^{1/2}$. The QPSD technique developed here could also be applied in such measurements, and hence can be extended to the high frequency range as mentioned by the referee. The challenge that we may encounter for our sensors while measuring such a broad range of frequencies would be to apply a bias field to cover the microwave frequency band.

In this manuscript, we envisage that the developed NV magnetometry for low frequency signal sensing protocols can have applications for telecommunications in challenging environments. On the other hand, for high frequency signals, conventional electronic systems are very sensitive, compact and cheap that nowadays quantum sensors can hardly surpass. We have included the discussion in the **Supplementary Note 2**, (please see the changes we mentioned in **R1-2**).

Reviewer #2:

Thank you very much for comments and suggestions on our manuscript! The following are the point-by-point responses and revisions to the manuscript.

#2-1: “Authors make good reference to adaptive/nonadaptive schemes for quantum phase measurement with improved LDR. Note that while Ref [24] is an adaptive scheme, Ref [23] uses a nonadaptive scheme so the sentence in 3-52 should be corrected accordingly.”

R2-1: Thank you for pointing out the unclarity of the reviewing and citing of two references in the manuscript. We do notice the differences and now have revised the sentence accordingly. See as the colored text in **line 56**. The reference numbers have been changed due to the new references.

#2-2: “I would like to bring authors attention to another paper where nonadaptive quantum phase estimation scheme was extended to detect ac magnetic fields: <https://doi.org/10.1103/PhysRevB.88.220410>. As the above work also demonstrate ac magnetometry with quantum phase sensing with improved LDR, it is indeed relevant to this manuscript so better cite it for completeness.”

R2-2: Thank you very much for bringing us attention to this highly relevant paper! We have now added the review and citation of this work in introduction, in **line 54-58**. We have now modified the text in the introduction highlighting this reference and describing its difference to the current work (in **line 58-61**).

#2-3:

- **Fig1(b) caption should be corrected to “..it can be understood..” .**
- **Fig4(b) captions says “..light blue corresponding to the right y-axis..” but there is no right y-axis.**
- **Fig4(d) better have legends to indicate what what red and black curves are.**
- **18-377 should be corrected to “.. pulses are fed ..”**

R2-3: Many thanks for finding out these corrections in our writing. We have corrected them accordingly. The correction of 18-377 is now at **line 391**.

#2-4: “One important data processing step for recognizing the systematic noise is explained in the end of Page 18 (18-392 onwards). It would be informative to see, perhaps in the supplementary, how the spectra looks like before and after applying this filtering method.”

R2-4: Thank you for the comment. We agree to the referees’ comment and have included additional text and figure in supplements. Please refer to our **Supplementary Note 8**, with **Fig. S3**. Here, we detailed the steps of the spectrum analysis. Figure. S3 shows the comparison of the original acquired spectra in the two steps measurement to the spectrum after the filtering. This part is cited in the main text at **line 303-304**, for clarity.

Reviewer #3:

Thank you very much for the very detailed review and valuable comments to our manuscript. We especially appreciate the insights the reviewer brings to us through his comments. We have tried our best to revise the text to make the statements more precise and clear in the claims and the description of the work. The following are the point-by-point responses and revisions to the manuscript.

Under comments: Audio sensing

#3-1: While the increase in dynamic range technique is interesting and useful to the field, the central message of the paper of audio field sensing as presented is misleading. The paper and supplementary files give the impression of actual audio sensing, while audio field signals were never measured in real time. The audio signal was classically preprocessed (compressed from 4 kHz to 200 Hz) and subsequently broadcast at this lower bandwidth in order to make the signal detectable by the NV diamond sensor. The NV-diamond-received-signal needed to be post processed (averaging sequential points in the time domain) in order to reconstruct an audio signal.

R3-1: Thank you very much for pointing out the possible misleading description in the manuscript. **Introduction** is revised to avoid the misleading of the main message in this paper.

The main message we want to show in the manuscript is a distortion-free sensing scheme that not only has extended linear dynamic range, but also the capability of resolving signal frequency. In the section **Results**, we use the first two subsections to elaborate the two features, and use the third subsection to demonstrate

the distortion-free signal detection, as shown in Fig. 4(a) and (b) where we see the detection of phase change and amplitude/frequency change. We also addressed the capability of detecting signals at frequency higher than a few kHz, rather than the near dc signals that OPMs can measure with an ultra-high sensitivity.

Regarding arguments for the audio signal sensing, we have revised **Supplementary Note 7** (previous Note 5) by adding a paragraph that discuss the bandwidth limit. To avoid misleading languages, we revised the **Introduction** section in the main text at **line 86-87**, as well as the paragraph from **line 325** in **Results C**.

The idea of detecting the electromagnetic signals at audio frequency band without distortions by taking the advantage of high dynamic range is exemplified here through two simple examples. We have only demonstrated a receiver for low-frequency broadcasting rather than a real-time communication system. We agree that pre-processing and post-processing is needed in this regards, which makes it not a real-time receiver. This has been addressed and presented as the section **Methods D**. We note that there are applications in which the time-lag is not that critical. For example, one potential and interesting application would be wireless communication (which uses radio signals at low frequency such as audio band) in challenging environments e.g. underwater and in underground. The trade-off is that signals with lower frequency has less reduction but a lower rate. Nevertheless, it is always expected to have a better system for immediate communication. In this case, we should discuss the limits of the current demonstrated setup, which were missing in the manuscript and supplements. We add the last paragraph highlighted in **Supplementary Note 7** for the discussion of the bandwidth issue. We also have revised the text at **line 354-356** to avoid misleading. Sentences are added to explain the application feasibility at **line 356-358**.

#3-2: The comparison of audio measurement to existing techniques is quite minimal, starting on line 58. This description incorrectly portrays quantum-based Rydberg sensors as requiring an antenna. Since they do not require an antenna, a comparison of sensitivity of the two, perhaps for a propagating EM field would be useful. Similarly, this could be extended to antenna.

R3-2: Thank you for pointing out the insufficient discussion and the incorrect description in the introduction text. First, we would like to elaborate the discussion in supplements with the comparison of the existing techniques (see in **Supplementary Note 2**). Please see also the response to **#1-2**. We have revised the **Introduction** of the maintext **line 62-77** addressing the following points:

i) The motivation of our work from the application side is that such advances in quantum magnetometers may benefit fields e.g. wireless communication in challenging environment, which requires detection of low frequency electromagnetic signals without distortion.

ii) One advantage of using magnetometers for such application is that, even though the sensitivity of the sensor can be not sufficient at the moment, flux concentrators can boost the sensitivity while keeping a compact size regardless the wavelength of signals.

#3-3: A stronger point, which would need analysis, could be that Rydberg sensors could also benefit from the QPSD higher dynamic range and extended bandwidth.

R3-3: Thank you for raising this point. We would be very happy to see the techniques being applied to other quantum systems. Regarding Rydberg sensors, they are based on electromagnetically induced transparency rather than using spin dynamics. Therefore, QPSD is not applicable to Rydberg sensors. However, for other types of quantum sensors that use similar interferometry method, this technique can be implemented to eliminate the sensing ambiguity for a higher dynamic range signal. We are delighted to make this clearly conveyed in the text, as the revised sentence in **Discussions** in **line 342-344**.

Under comments: Dynamic range

#3-4: As dynamic range is a significant point of the manuscript, an intuitive description (beyond lines 128-130) of how this technique works would be helpful. In addition, a quantitative method comparison table of the traditional (Ramsey, frequency locking, other techniques) vs. QPSD technique is needed. This table could show trades in sensitivity, dynamic range, frequency range of applicability, for example, putting the work into context.

R3-4: Thank you for the valuable comments. We revise the text following your suggestion.

1) We hope that we made the description of the technique clearer by revising the first paragraph in this section, in which we highlighted the text from **line 92 to line 102**. In **line 104-108**, we point out how the linear dynamic range can be extended.

2) Thank you so much for the suggestion. We prefer to add the comparison table into the supplements, see the new **Supplementary Note 1**. The **Table S1** is summarized according to your suggestion. While in the manuscript, we revise text in the **Introduction** at **line 48 – 61** for the comparison.

#3-5: Figure 2 shows an experimental comparison of QPSD to Ramsey sensing sequences. Significantly, what the limit of QPSD, theoretically? The manuscript seems to be motivated by

practicality of pulse sequence for increasing dynamic range and bandwidth. Translating $-\pi$ to π to frequency could potentially assist in relaying this description. Is figure 2C referring to Ramsey readout out? These plots should be compared theoretical curves.

R3-5: Thank you for the questions.

1) Yes, one theoretical limit of the QPSD scheme in dynamic range is phase wrapping over a cycle of 2π . Indeed, technically by adjusting the phase factor of the demodulation reference in the lock-in detection can relay the extension. We manually did this in experiments.

At **line 104-108**, we discuss the limit and the technical solution. The issue is also now pointed out in the comparison table in supplements (see **Supplementary Note 1** and **Table S1**).

2) Yes, the plots are based on results of Ramsey experiments. As for QPSD scheme, the phase readout is expected to be a constant because we applied a calibration signal keeping the same amplitude. The line is plotted to indicate the result in Fig. 2c. The experimental error could come from the long term drift of the bias field. We kept the same MW frequencies in all the measurements. All the data points are below 1 because we normalize the plotting data with the maximal readout.

As for the fluorescence readout, we revise the text at **line 182-187**. We add the reference [43], which is one of our recent publication and in which Fig. 2(c) and the supplements discuss the fluorescence readout dependency on the sampling frequency. We do not have a general model for a theoretical curve because the spin polarization varies also with the external fields when the polarization rate is low. We use experiments to get the polarization rates for calculating fluorescence contrasts, which is not a theory curve.

Under comments: Substantiation of claims

#3-6: In general, throughout the paper, the claims need to be substantiated. The abstract notes experimental and theoretical investigations reaching 98 dB linear dynamic range, 31 pT/rtHz sensitivity, and arbitrary frequency/phase resolution (which would be limited by a master clock). None of these numbers seem to be substantiated in the manuscript. The sensitivity value is stated and the dynamic range is inferred from scale factor. Line 315 notes that a sensitivity is achieved, but there is no data to show how this is obtained.

R3-6: Thank you very much for pointing out the issue. We agree that additional details on how the values are obtained should be added. We also agree on your **Comment 19** that the sensitivity is not the focus of the paper. So we have revised the **abstract** and the text in the manuscript. **Subfigures** are added in Fig. 2

(Fig. 2c and d) together with the section **Methods B, line 413-430** to give substantiation of the values. Discussion on frequency resolution is added in the end of section **Results B, line 277-279**.

- **Sensitivity:** Calibration results and noise measurements are added in Fig. 2. Thank you for your **comment 7** and **comment 12** that remind us to examine the values.

We notice discrepancies in previous calculation that the noise level η_{phase} rather than the standard deviation σ_{phase} should be used for characterizing the sensitivity. σ_{phase} should be used for characterizing the frequency estimation. We verify the QPSD sensitivity value with the fluorescence readout sensitivity value presented in **line 175** and **line 426-430** confirming the $\sqrt{2}$ times difference. We also revise the **Supplementary Note 5** (previous note 3) for details.

We note another mistake in the previous text is that the experiments were done together with measurements in fig.2 that uses $T_\phi = 12.5 \mu s$ in the Hahn-echo sequence. Although sensitivity with $T_\phi = 50 \mu s$ could be higher, we didn't measure it because we focused on the distortion-free reconstruction of signals. **Equation (6)** is revised, and the Hahn-echo measured sensitivity is in proportional to $1/(T_\phi e^{-(T_\phi/T_2)^p})$. With $T_2 = 200 \mu s$, it is calculated that the sensitivity improves by about 3 times if T_ϕ is changed from $12.5 \mu s$ to $50 \mu s$ ($p \approx 0.5$). Nevertheless, we prefer to keep the measurements we have done and rearrange the structure by adding the sensitivity estimation section in **Methods B**.

- **Dynamic range:** Statement of dynamic range is revised and moved to **line 426-430**, as they are calculated based on the estimated sensitivity. We note that the calculated dynamic range is specified to measurement without feedback or an algorithm for addressing the phase wrapping.

- **Frequency resolution:** Although high frequency resolution is another feature of the technique described in the manuscript due to the combination of heterodyne readout, it has been sufficiently discussed in other literatures (Ref. [24, 25]). It is discussed now in **line 277-279**.

#3-7: The manuscript notes that the experiment is performed under low contrast (how low?), and under these conditions, it would be difficult/impossible to reach shot noise limit. Details on experimental parameters are needed. Does this "achieved" sensitivity agree with the theoretical prediction of $\sqrt{2}$ less than traditional methods? Give sensitivity for both Ramsey and Spin Echo with and without QPSD?

R3-7: Thank you for the very important questions. **Equation (5) and (6)** are revised by adding the decoherence term that reduces detected contrast when T_ϕ is different. We reported the low contrast $C < 0.2\%$

case in the Discussions in **line 350-352**, and the value is used in **Supplementary Note 5** to estimate the shot noise limit (**below Equation S22**).

In the new **Fig. 2c** and **d**, and **line 172-176**, and **Methods B**, we present the calibration of the sensitivities by using QPSD readout and the fluorescence readout. The reported value shows the $\sqrt{2}$ difference.

The sensitivity calibration is presented based on Hahn-echo measurements for both QPSD readout and fluorescence readout. As for sensitivity of Ramsey, we did not present the calibration data because there is near dc magnetic noise from the 3D coils that we used to generate B_0 field. The measured noise only shows the quality of the current supplies and cannot be used for analysis of shot-noise limit.

#3-8: Similarly, Line 316 denotes the dynamic range without corroboration. How does the achieved compare with existing techniques? The dynamic range appears to be projected rather than measured.

R3-8: Thank you for the question. As responded in **R3-6**, text in **line 426-430** is revised to have a better discussion on dynamic range. For comparison with other existing techniques, we prepare **Supplementary Note 1** with **Table S1**, in which we make comparison of the features of techniques mentioned in the manuscript.

Since we did not address the phase wrapping in experiment, we calculate the dynamic range from the natural phase limit of 2π . However, the issue can be addressed and we outlook it in the text now. Compared to work that only address high dynamic range, in this work we focus more on resolving arbitrary signals.

#3-9: Line 25 says that a narrow NV linewidth limits the full dynamic range of a sensor. This is only true for open-loop sensor configurations in which the resonance is not tracked. The authors clarify this point later in the manuscript, it should be noted the first time the technique is mentioned.

R3-9: Thank you for the suggestion. We now add the point in **line 25**, and revise the paragraph from **line 48** and also add **Supplementary Note 1** for comparison between the techniques.

We note that tracking the resonance still has technical challenges in both ways, i.e. compensating field and tracking resonant frequencies. On the hand, the QPSD can also use close-loop to further extend the dynamic range beyond the phase cycle limit of 2π . Unlike classical method, in QPSD scheme we only need to adjust the reference phase that used for demodulation to address the phase wrapping. The point is now added in **line 104-108**.

#3-10: There is also a note that the frequency tracking method is only valid when the frequency to be sensed is less than sampling frequency (line 50). While it does not affect the proof-of-principle result, Figure 2 data are taken in a regime with 46 Hz signals with kHz sampling rates.

R3-10: The sentence is revised in **line 52-54**. We want to make a point that the interferometric scheme used for ac sensing archive all the information, i.e. frequency, amplitude and phase, into one physical quantity, i.e. quantum phase. This makes it difficult to use a feedback loop to do adaptive measurements.

Yes, we measure heterodyne frequency to determine the frequency of signal to be measured. The measurement sequence determines a reference frequency for detection, and in the case of Fig. 2(b) we use $T_\phi = 12.5 \mu s$ to measure signals around 80 kHz. Therefore, when we measure the signal at 80.046 kHz, the readout shows the signal at 46 Hz.

#3-11: Also, the tie to telecommunications is not clear. In addition, the language in this section is not precise, with exaggerated statements. Line 27 notes, “massive limitation on sensitivity”. And lines 31-32 note that operating within linear dynamic range of a sensor can be crucial for reconstructing unknown signals, whereas many antenna today have limited dynamic range and solve this issue through post processing and calibration (although it would be better not to have to do so).

R3-11: Thank you for the comment. We have revised the exaggerative statements in the text. The paragraph is revised in introduction **line 62-77**. We try to make a point that the technique directly shows a potential in application of telecommunication in challenging environment, and more importantly, the technique path way to reconstruct signals in multiple frequency bands with decent sensitivity and compactness. In **Supplementary Note 2** and **Table S2**, we add details and compare the existing techniques for detecting oscillating fields. We remark that conventional antennas have extraordinary performance at high frequencies but has a disadvantage in size at low frequencies. We also hope the technique could inspire other applications that use other quantum systems with interferometric schemes.

Under comments: Experimental methods

#3-12: How scale factor is determined/verified to apparatus description including light collection/readout.

R3-12: Thank you for the question. We add details in the **Supplementary Note 5** (first paragraph).

We remark that the scalar factor denotes the relationship between the quantity to be measured and the readout, which in the case of QPSD readout is $k = \frac{\delta\phi}{\delta B} = \gamma_e G(\omega)$. It does not include the light collection/readout. Light collection/readout only affects the sensitivity. In equation (6), the detected photon counts can be expressed as $\mathcal{N} = \epsilon N_{NV} n_{avg}$, where ϵ is the collection efficiency, N_{NV} is the evolved number of NV centers, and n_{avg} is the average number of emitted photons per NV center.

#3-13: While laser type is given, Sprout G (nominally 10 W) is limited to 80 mW at the diamond.

R3-13: Thank you for the question. Yes, Sprout G series has a high output power. Though, we mainly take the advantage of low laser noise (<0.03%). Since we do not focus on a high sensitivity, we lowered the RF driving level of the AOM (results in diffraction rate ~10%) to have a low laser power. By this, we can reduce the heating induced drifts in the optical path.

#3-14: Microwaves are delivered through a dielectric resonator without details of delivery, coupling, NV axis orientation; does the resonator effect the bandwidth of signals?

R3-14: Thank you for the comment. We add a sentence in line 392-393.

The magnetic component of the MW is coupled vertically to the (111) direction of the diamond. It is possible that a dielectric resonator can effect signals to be detected. However, we are detecting signals in the kHz range while the resonator is designed for GHz. We also note that the MW field can be sent to the diamond by simply using loop antennas.

#3-15: If test fields are applied by a loop, how does the polarization of the applied test field interact with the NV class being used?

R3-15: We measure the field projected on one NV orientation. Polarization is only treated as the dynamics of the field amplitude in this case. Vector magnetometry can be achieved by using the four NV classes. Polarization of signals can be reconstructed if the vector magnetometry is implanted.

#3-16: How is the florescence signal detected?

R-16: Text is added in **line 387-388**. We use a compound parabolic lens to collect the fluorescence and detect them with a large area photodiode.

#3-17: The diamond is surrounded by 3D coils. Are they aligned such that all NV classes are used or one is separated out?

R-17: We use the 3D coils to generate a bias field along the (111) direction, i.e. one of the NV orientations is separated out. Text is added in **line 384-385**.

#3-18: How are the magnetic fields of the test loop calibrated?

R3-18: Thank you for the question. The calibration method is now added in **Methods B** in **line 406-413**,

We calibrate the test loop by using the ODMR spectrum of the NV sensor. The splitting of the resonant lines depends only on the external field and changes with electron gyromagnetic ratio as the coefficient.

Comments under: “Significant points...”

#3-19: Abstract leads with metrology and sensitivity, when neither is the focus of the paper.

R3-19: Thank you very much for the suggestion. We revised the abstract accordingly.

#3-20: Flux concentrators direct more signal at the diamond. This does not fundamentally change the sensitivity of the device, it directs more signal toward the sensing medium at the expense of long term stability, vector accuracy, etc.

R3-20: Thank you for the comment. We agree that flux concentrators do not fundamentally change the sensitivity of the sensor. Text is revised in **line 358-359** to point out the concern of losing vector property.

Here we note that flux concentrators can be seen as antennas that are used in conventional systems. Using flux concentrators on diamonds can improve SNR more significant than other type of quantum sensors due to the high volume-normalized sensitivity. Though it is not shown in this work, the advantage of using flux concentrator is well discussed in **Ref. [39], [40]**. We prefer to keep using the phrase “improve sensitivity” by seeing the sensor+flux concentrator as one device in consistency to literatures.

#3-21: NV capability is used interchangeably with quantum sensors. Note that to date sensors such as SQUIDS and vapor cells have far exceeded the sensitivity of NV diamond. NV sensors are not competitive with these devices in the manner portrayed.

R3-21: Thank you for the comment. In this manuscript, we do describe the technique with NV magnetometer interchangeably with quantum sensors, and we want to make the point that the technique is developed for distortion-free sensing. We hope it can be used for not only NV magnetometer, but also other quantum sensors that use interferometric schemes or at least for NV sensors to measure other quantities. We revise the **Abstract** and the introduction text **line 31-34**.

The second point is also true that NV sensors are not as sensitive as SQUIDS and vapor cell magnetometers. Nevertheless, it still has features that both SQUIDS and vapor cell sensors that cannot achieve. From **line 35 to 42** we present the motivation of using NV centers and compare it with OPMs. First of all, the NV center is a room-temperature based platform, while SQUIDS requires cryogenic. It makes SQUIDS not competitive regarding compactness and cost. Secondly, although OPMs are integrated well, they have a drawback in bandwidth. Most of the OPMs can only acquire high sensitivity near DC, and the sensitivity is not competitive compared to NV sensors when the signal frequency goes up to kHz-MHz range. Besides, using of flux concentration can further improve the sensitivity to the OPM level while the device still has a centimeter size.

#3-22: Audio signal could be directly sampled by the NV sensor with no loss in sensitivity with a long-coherence-time diamond.

R3-22: Thank you for the comment. Yes, by using NV sensor with a longer coherence time, one might directly detect audio signals at frequency of a few kHz, or detect signals at the same frequency presented in this work but with a higher sensitivity. We note that T_2 of the diamond in this work is 200 μs (Ref. [11]), which corresponds to a detectable frequency of 5 kHz. According to the heterodyne readout technique described in this work, the detectable signal frequency can be as low as 2.5 kHz with the same sensitivity when detecting 5 kHz signal. Although in the paper we try to only focus on presenting the techniques, it would be clearer if we put this argument into discussion. Therefore, we revise text in the discussion section by adding a sentence at **line 354-358**.

#3-23: Language tends to overstate without substantiation ex. “massive”, “crucial”, “remarkably” (in Results section B).

R3-23: Thank you for point out the issue. We remove the exaggerative words as you suggest.

#3-24: There are quite a few typos in the manuscript: (e.g. Tekreonix on line 376 or frequency offset, phase of MW2, and Rabi frequency, respectively on line 110 is in wrong order). Many variables, lines in plots, etc are not defined. Figure1c: In the pulse sequence diagram, what do the green and brown colors signify? Much of the information in figure descriptions could be in the text itself.

R3-24: Thank you for pointing out the issues. We revise the text accordingly:

1) We correct the typos that you give here, and go through the text searching for other typos.

- “Tektronix” on **line 390**.
- We exchange the order of Ω_2 and β on **line 118**.
- Fig. 4b caption revised
- **Line 391:** “pulses are fed”

2) We go through the variables and figures in the manuscript, and correct as follows

- **Fig. 1c:** the green stands for the laser applied at the detection windows, and the red color represents the detected fluorescence, which drops at beginning due to the superposition and reinitialized with the green laser illumination. We revise the captions to make the descriptions clearer.
- **Fig. 4d:** missing legends added.
- **Line 114:** $f_0 = \omega_0/2\pi$.
- **Line 222-223:** explain variable “m” and “ ω ”.
- **Line 241:** $B_H(t)$ is the heterodyne readout.

3) We revise all the captions by removing the repeated description that also appears in the main text.

- **Fig. 1:** The caption is revised by removing the repetitive text and adding the explanations of the symbols shown in the figure.
- **Fig. 2:** We add two subfigures and the captions are revised accordingly.
- **Fig. 3-4:** The captions are revised by removing the repetitive sentences that also in the manuscript.

#3-25: Statements such as in line 41 “The NV magnetometry has been performed under bias fields ranging from zero-field to a few Tesla” need citation.

R3-25: Thank you for the comment. We add the citation to the statement, now in **line 40**.

#3-26: For equation (3), the authors should point to Supplementary note 1 since that is where the expression is derived. Also, in S3, the evolution operator acts on the wavefunction, not the other way around. The exponential and ket should be switched.

R3-26: Thank you for point out the issues.

1) We add the reference to the supplements at equation (3) in **line 122**.

2) We correct the equation S3 in the supplements.

#3-27: Why does the theoretical filter function for the CPMG-2 sequence not really match the measurement? It does for the Hahn-echo sequence.

R3-27: Thank you for the question. The discrepancy mainly happens at low frequencies. The theory matches the CPMG-2 measurement well at high frequencies. We attribute this discrepancy to the pulse errors in experiments. Unlike theory in which we calculate neglecting the pulse width, the experimentally applied driving field pulses have finite pulse width and can also have errors to π or $\pi/2$ pulses in practice. The discrepancy is now talked about in **line 251-252**.

Finally, we would like to thank the referees again for their time and careful reading our manuscript. All this rigorous comments are well appreciated as it made us improve and clarify many parts of our manuscript in this resubmission. We are hopeful that all the changes amended are satisfactory.

REVIEWERS' COMMENTS

Reviewer #1 (Remarks to the Author):

The authors have addressed my comments in very detail, thus I would be happy to recommend its publication in the current form.

Reviewer #2 (Remarks to the Author):

This work uses Nitrogen-Vacancy (NV) ensemble to demonstrate quantitative probing of low frequency weak ac magnetic fields, proving its potential applications in low-frequency telecommunication under challenging environments. Authors have adequately responded to previously raised comments by reviewers and the manuscript is clearly improved in its clarity. Therefore I would extend my recommendation for publication of this work in Nature Communication.

Dr. Chen Zhang
3rd Institute of Physics,
University of Stuttgart
Allmandring 13,
70569 Stuttgart, Germany

Dear Reviewers,

Since there is no further comment to our manuscript, hereby, we would like to appreciate your reviewing work. Thank you very much for all of your comments to our manuscript! We are very glad that you are satisfied with our revised manuscript!

Best regards,

Chen